# Impact of trainability on telomere dynamics of pet dogs (*Canis lupus familiaris*): An explorative study in aging dogs

Julia Weixlbraun[1], Durga Chapagain[2], Jessica Svea Cornils[3], Steve Smith[4], Franz Schwarzenberger[1‡], Franz Hoelzl [4‡]*

1 Unit of Physiology, Pathophysiology and Experimental Endocrinology, University of Veterinary Medicine, Vienna, Austria, 2 Messerli Research Institute, Department of Interdisciplinary Life Sciences, University of Veterinary Medicine, Vienna, Austria, 3 Research Institute of Wildlife Ecology, Department of Interdisciplinary Life Sciences, University of Veterinary Medicine, Vienna, Austria, 4 Konrad-Lorenz-Institute of Ethology, Department of Interdisciplinary Life Sciences, University of Veterinary Medicine, Vienna, Austria

‡ FS and FH shared last authors on this work.

* glis.glis85@gmail.com

**Data Availability Statement:** All relevant data are within the manuscript and its Supporting Information files.

## Abstract

This research studied the impact of various factors (including social and physiological parameters) on telomere dynamics in pet dogs. Telomeres, essential for maintaining genomic integrity, undergo shortening with each cell division, leading to cellular senescence. Previous studies in humans have linked cognitive and social factors with telomere dynamics but in animals, such associations remain understudied. This study is based on a previous study, where behavioral and cognitive changes in aging pet dogs were investigated. Together with standard variables (sex, age, body weight, diet), behavioral predictors that were assessed in the "Modified Vienna Canine Cognitive Battery" were used. This study aimed to investigate the influence of these factors on telomere dynamics in aging pet dogs. The relative telomere length of 63 dogs was measured, using a qPCR method and a model selection approach was applied to assess which variables can explain the found telomere patterns. Results revealed a strong association of the behavioral factor called trainability and telomere change. Trainability was the best predictor for telomere change over time and was the only predictor having a relative variable importance (RVI) above 0.7. This finding suggests that higher trainability positively affects telomere dynamics in aging dogs and factors like age, sex, diet, and other cognitive parameters are less important. The study sheds light on the potential role of cognitive factors in canine aging and offers insights into improving the quality of life for aging dogs, but further research is needed to comprehensively understand the interplay between behavior, cognition, and telomere dynamics in dogs.

## Introduction

In Mammals, telomeres are the end caps of linear chromosomes and contain many tandem repeats of the sequence 5'-TTAGGG-3'. Together with telomere-associated proteins (the

**Funding:** The author(s) received no specific funding for this work.

**Competing interests:** The authors have declared that no competing interests exist.

shelterin complex), they prevent the degradation of the coding DNA [1]. Telomere shortening (TS) occurs with each cell division due to the end-replication problem in mitosis [2, 3]. In addition, oxidative stress erodes telomeres [4, 5]. Advanced TS can lead to loss of protective function of telomeres and induce replicative senescence or apoptosis of cells [6, 7]. These factors opened an ongoing debate about whether telomeres can be considered a biomarker for aging. Studies suggest that this is true in some cases and that, in many species, telomeres shorten throughout an individual's lifetime [8–11]. Other studies suggest however, that the correlation between telomere length and age is not always linear and negative [12, 13]. This could be due to the possibility that telomeres can be elongated by the activity of the enzyme telomerase or by alternative lengthening of telomeres [14–18].

Even though the underlying functions of changes in telomere length in humans are well understood, the complex factors influencing telomere dynamics in animals require further study. A majority of studies have focused on the relationship of telomeres and growth rate, life history strategies, nutritional interventions, and other physiological parameters [10, 12, 19], however less is known about the influence of social and cognitive factors on telomere dynamics. While a study done in African gray parrots has described how social isolation shortens telomeres [20], research done with children have shown shortening of telomeres when exposed to a disadvantaged social environment and lengthening of telomeres when living in an advantaged environment [21].

Apart from the direct influence of social effects on telomeres, it has been shown that physical, or the combination of physical and cognitive training have a positive effect on health. Furthermore, physical activity, nutrition and cognitive training can reduce the rate of age-dependent cognitive decline and promote physical and cognitive health [22–37]. However, little is known about the influence of these parameters on telomere dynamics in non-human mammals like domestic dogs (*Canis lupus familiaris*). Dogs are considered to be intelligent [38–41] and due to thousands of years of coevolution and domestication, have acquired social skills that enable them to live in very close relationships with humans [41]. They were one of the first species to be domesticated, sharing immediate and daily environments and lifestyles [42–44].

Dogs have similar telomere length and telomerase activity as humans. Although dogs' telomeres shorten approximately ten times faster than telomeres of humans, this is consistent with the difference in life expectancy between humans and dogs [43]. Because of their limited life expectancy, their telomeres are easier to study throughout their lives. The lifespan of dogs varies greatly due to their different breeds, and different breeds also show differences in behavior, genetics and mean telomere length [38, 43, 45–47].

A few studies describe telomere dynamics in dogs: Some described the dynamics of telomere length and telomerase activity in canine cancer cells [48–50], others validated the use of oral swabs for telomere length assessment in dogs [51]. [52] tested leukocyte telomere dynamics in dogs.

This study measured telomere change in a group of aging pet dogs, already studied by [53] in a study entitled "Behavioral and cognitive changes in aged pet dogs: No effects of an enriched diet and lifelong training". [53] examined the effects of age, an enriched diet, and lifelong training on various behavioral and cognitive measures in 119 pet dogs (> 6 years of age). The dogs were fed either an enriched diet or a control diet for one year. Lifelong training was calculated using a questionnaire in which owners entered their dogs' training experiences. Before starting the diet and after one year of dietary treatment, they were tested in the Modified Vienna Canine Cognitive Battery (MVCCB). The MVCCB, developed by the Clever Dog Lab at the University of Veterinary Medicine, is a test battery that assesses dogs' cognitive abilities in a controlled environment, focusing on aging and behavior changes. It includes tests on

memory, problem-solving and sociability, helping to understand how dogs perceive their world. This tool aids to identify cognitive deficits and compares abilities across breeds to enhance insights into canine cognition.

The MVCCB consisted of 11 subtests designed to assess correlated individual differences in a series of tasks measuring general, social and physical cognition and human-animal interactions. This resulted in six final factors: problem solving, trainability, sociability, boldness, activity-independence and dependency. Problem solving, sociability, boldness, and dependency showed a linear decrease with age, suggesting that the MVCCB can be used as a tool to measure behavioral and cognitive decline in older domestic dogs. An enriched diet and lifelong training had no effect on these factors. For more details, see [53].

For all the aforementioned reasons, and because dogs represent a promising model for genetic analyses related to telomere biology, we conducted this explorative study to determine how a dog's biological and cognitive differences may affect telomere dynamics, and thus, have potential implications for quality of life and lifespan.

## Material and methods

### a) Ethical statement

The experiments of the study by [53] were approved by the institutional ethics and animal welfare committee at the University of Veterinary Medicine Vienna (Protocol number: 05/03/97/2014), and the Austrian Federal Ministry of Science, Research and Economy (BMWFW; permit number: BMWFW-68.205/0151-WF/V//3b/2014). All additional experimental protocols and buccal swabs were approved (protocol number: GZ BMWF-68.205/1035-V/3b/2014) and carried out in accordance with the approved guidelines by the Law for Animal Experiments § 8ff (Tierversuchsgesetz–TVG). The dog owners signed a form of consent before participating in the study.

### b) Animal subjects

Out of the 119 pet dogs in the study by [53], 63 were included in this study. This reduced number was due to various interfering factors such as death, exclusion of the study, lack of samples, missing data points or failed PCR during the data curation. The 63 dogs contained 25 mixed breeds and 38 pure breeds of the FCI-group number 1 (19 individuals), number 8 (6 individuals) and the FCI-group numbers 2–7 and 9, which were merged to a group of 13 individuals (diff for different). The Fédération Cynologique Internationale (FCI), or International Canine Federation, is the largest global organization of national kennel clubs and purebred registries. It classifies recognized dog breeds into ten groups based on characteristics such as appearance and function (http://www.fci.be/en/; for more information see S2 Table).

There were 37 females and 26 males. Among the study animals, 18 male and 27 female dogs were neutered, these individuals were categorized as male or female despite their neutered status. Their age (at first sampling point) ranged from 6.2 years to 13.4 years, with the mean age being 9.5 years. Average body mass of all dogs was 20.4 kg, ranging from 8.0 to 37.7 kg. Prior to participating in the study, all dogs were thoroughly examined by Veterinarians at the Clinical Unit of Internal Medicine of Small Animals of Vetmeduni Vienna. Additionally, standard blood cell and serum biochemistry tests were carried out in the beginning of the study to ensure that the dogs were healthy and qualified to participate in the study.

During recruitment, owners filled out an extensive demographic questionnaire detailing their dog's lifelong training experiences on 13 different types of training. For each type of training, a score was determined for each dog based on their past and current training participation. A lifelong training score was then obtained by summing up all the collected scores,

ranging in our study from 0–31. The average lifelong training score of the 63 dogs was 10.9. For details see [53].

In the study by [53], the 119 dogs were divided into two groups matched for sex, age, life-long training score, breed and weight, and each group received either the control or test diet for a period of 1 year. The control and the test diets were similar in composition except that the test diet was also enriched in antioxidants (Vitamin C, Vitamin E, Polyphenols), omega 3 fatty acids (DHA, Docosahexaenoic acid), phospholipids (Phosphatidylserine), and a higher level of tryptophan. Nutritive intake of each dog was calculated separately based on the weight, age and body condition score. Dog owners were given food bags every month and instructed to exclusively feed their dogs with the given diet, with no more than 10% of other treats. On the days when the dogs came for training at the Clever Dog Lab, dogs were provided with low caloric training treats and did not receive any other treats from their owners. Neither the experimenter nor the owner knew which were the control and the enriched diet until all analyses had been conducted [53, 54]. Before starting the diet and after one year of dietary treatment, the dogs were tested in the MVCCB consisting of 11 subtests to examine correlated individual differences in a set of standardized tasks measuring general, social and physical cognition and related behaviors. For further information see [53].

## c) Relative telomere length

This study used buccal cells as a source of genomic DNA. This tissue type was chosen because 1) it is considered a tissue type with stable cell-turn-over rates [51] and 2) sample collection is minimally invasive. The samples were collected in collaboration with the Clever Dog Lab and University of Veterinary Medicine Vienna. The buccal mucosa samples to assess initial relative telomere length (RTL) were collected during the first veterinary examination appointment at the Clever Dog Lab (directly before the start of the dietary treatment). The second sampling was performed at the last training at the Clever Dog Lab (directly after the dietary experiment had finished). Before sampling, the cheeks were checked for any remaining food debris.

The buccal mucosa samples were taken with Gynobrush®-Brushes (Heinz Herenz Medizinalbedarf, Germany). The brush was twirled for about 15 seconds in the inner cheek. After the cells were collected, the brush was put in a 1.5 ml Eppendorf-tube filled with 1 ml of BC-buffer (100 ml: EDTA (0.5M) 20 ml, NaCl (5M) 0.4 ml, TrisHCl (1M) 1 ml, H2O 78.6 ml. pH 7.0 (adjusted with HCl)). The head of the brush was cut off, the tube closed and stored vertically at 4–7 C° until the DNA extraction. DNA samples were extracted within 48 hours of sampling. Prior to the DNA extraction, tubes were centrifuged to collect the cell pellet at the bottom of the tube and the brushes were aseptically removed. The extraction was performed with the DNeasy Blood & Tissue Kit (Qiagen) according to the manufacturer's protocol. DNA concentration was measured with the NanoDrop 2000c (Spectrophotometer, Peqlab, Germany) and diluted to 10 ng/µl with nuclease free water. The DNA samples were stored at -80°C until sampling was completed. RTL was measured using the real-time PCR approach [11] adapted for pet dogs. As non-variable copy number (non-VCN) gene, we used the C7orf28b gene, described as non-VCN by [55]. Primer sequences for the non-VCN gene are 5′-GGG AAA CTC CAC AAG CAA TCA-3' (C7orf28b_3_F) and 5′-GAG CCC ATG GAG GAA ATC ATC-3' (C7orf28b_3_R), and telomeric primer sequences are 5′-CGG TTT GTT TGG GTT TGG GTT TGG GTT TGG GTT TGG GTT-3′ (Tel 1b) and 5′-GGC TTG CCT TAC CCT TAC CCT TAC CCT TAC CCT TAC CCT-3′ (Tel 2b). Telomere and non-VCN gene PCRs were carried out in separate runs with 20 ng DNA per reaction. Each telomere reaction contained 10 µl of GoTaq qPCR Mastermix (Cat.Nr. 6001; Promega), 600 nmol of forward and reverse primer and 200 µmol dNTPs. Each non-VCN reaction contained 10 µl of GoTaq

qPCR Mastermix and 300 nmol of forward and reverse primer (C7orf28b). To minimize pipetting errors, reactions were prepared using the Qiagility Robot (Qiagen). PCR cycling was performed on a Rotorgene Q instrument (Qiagen). PCR conditions for the telomere primers were 2 min at 95˚C followed by 40 cycles of 10 s at 95˚C, 20 s at 56˚C and 20 s at 72˚C. For C7orf28b, PCR conditions were 2 min at 95˚C followed by 40 cycles of 10 s at 95˚C, 20 s at 58˚C and 20 s at 72˚C. A final melting step was included in each run with the temperature ramping from 72 to 95˚C in 1˚C steps. All ratios of telomere to non-VCN gene were compared with a reference standard sample (A, RTL = 1), which was included in every run. A negative (no-template) control and a second standard sample (B, to evaluate inter-run variation) were also included in each run.

The intraclass correlation coefficient (ICC) was calculated within and between the runs, reflecting the degree of agreement and correlation [56]. ICC estimates and their 95% confidence intervals for sample triplicates were calculated in R Version (Version 4.2.3. [57]). Intra-rater ICC was calculated on all included data points based on a single-rating, absolute-agreement, 2-way mixed-effects model (ICC in library 'irr', [58]). Intra-assay ICC for telomere assay $C_t$ values was 0.996 ($p < 0.0001$, 95% (CI 0.995–0.997)) and for the non-VCN gene 0.994 ($p = < 0.0001$, 95% (CI 0.992–0.995)) showing a high level of reliability. Inter-assay ICC was calculated for standard samples A and B based on a mean score ($k = 3$), consistency, 2-way mixed effects model. Inter-rater ICC for telomere assay $C_t$ values was 0.998 ($p = < 0.0001$, 95% (CI 0.985 to 1)) and for non-VCN gene 0.971 ($p < 0.0004$, 95% (CI 0.781 to 1)) demonstrating an excellent level of reliability.

## d) Statistical analysis

The freely available software LinRegPCR (2012.0) [59, 60] was used for analysis of non-base-line-corrected raw qPCR data, exported from the instrument. RTL was calculated using a modified formula described previously in [61]. For details see also [62]. Statistical testing was carried out using R (Version 4.2.3, [57]) and the R package MUMIn [67]. For Fig 1 the packages ggplot2 [63] and ggeffects [64] were used. RTL change was computed by final RTL as percentage of initial RTL (POTC, percentage of telomere change). All predictors were tested for correlation [57]. The linear model calculated, testing the effect on RTL change, included POTC as the response variable and initial RTL, FCI-group, weight, sex, age at first sample date, diet, sociability, trainability, boldness, dependency, problem solving and activity-independence as predictor variables [57]. A detailed explanation of all used variables can be found in Table 1.

When using the changes of RTL between two time points the potential problem of 'regression to the mean' may arise. Simply spoken, it refers to the fact that an extreme value measured in a subject during a first trial is likely to lie closer to the mean in subsequent trials [65]. Therefore, an animal with longer initial RTL would likely lose more telomeric repeats than an animal with shorter initial RTL. If RTL shortening is constant, one would expect a negative effect of initial RTL on the reduction of RTL or vice versa. To overcome this potential problem, we use initial RTL as a covariate in the model. Such an approach has the advantage that all predictor variables are entered simultaneously, and the correct degrees of freedom are used (for details see [12]). In this study, the constancy of variance and the normality of errors were visually assessed using the built-in model checking tool in R [57].

To investigate which factors predicting the POTC best, we performed a model selection based on AICc (Akaike information criterion with a correction for small sample sizes). Model selection was performed on all possible sets of predictors, followed by model averaging. In addition, relative variable importance (RVI) of all variables in the model selection was

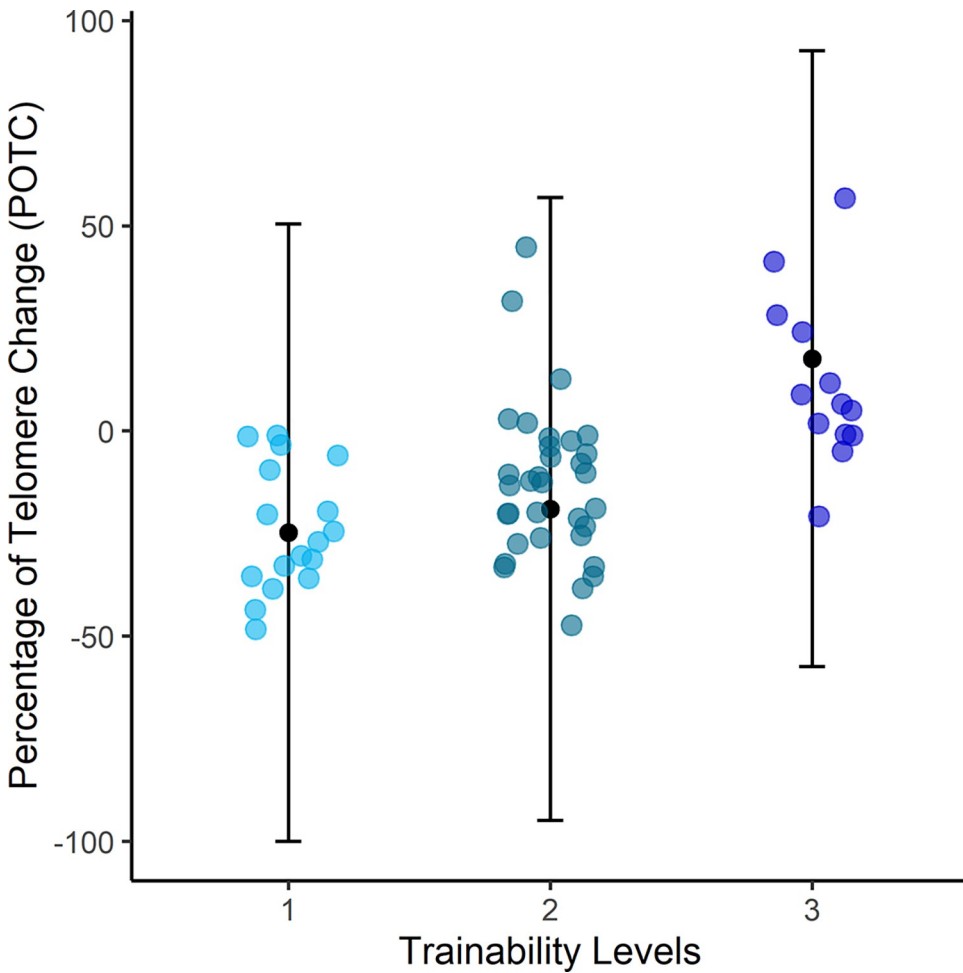

**Fig 1. Reproducibility of one run including 16 samples".** Run 1" is indicating the first and "Run 2" the second qPCR analysis of the identical samples, the black line shows the regression line of the points, the gray area shows the 95% confidence interval.

calculated [66]. RVI is setting the most important variable to 1.0 (or 100%) and ranks all other variables compared to the single best variable.

To test for reproducibility in addition to intra- and inter-class ICC, we repeated one run 3 months after the first attempt. Both telomere and non-VCN assay were set up, performed, and analyzed identically. The comparison of both attempts showed a very high reproducibility ($R^2$ = 0.93) and are shown in Fig 2.

## Results

In the six best models, within a delta of 2 AICc units of the lowest value in a best-models subset (package MUMIn; [67]), five variables were included in the best models (Table 2). The selection was trainability, age at first sampling date, diet, body mass and initial RTL, as the parameters that explain the POTC best. Therefore Table 3, which shows the Relative Variable Importance (RVI) (package MUMIn; [67]), only reports the RVI for these five variables. Trainability was found to be the only variable to be important, all other variables in the model are not informative predictors of POTC (Table 3). The influence of trainability on POTC showed

**Table 1. Variables used to build the initial model. Information in squared brackets indicates if the variable was entered numerically or categorical in the model.** The social parameters comprising the dogs' behavior in multiple situations. For more details on the composition of these social parameters see [53].

| | |
|---|---|
| POTC | "Percentage of telomere change"; final RTL as percentage of initial RTL; [numeric] |
| initial RTL | "Relative telomere length" at the first sampling date; [numeric] |
| FCI-group | Dog breeds grouped by the international breeding association "Fédération Cynologique Internationale (FCI)"; for details see S2 Table; [categorical] |
| weight | Body weight of the dogs in kilogram; [numeric] |
| sex | Female or male (neutered or not); [categorical] |
| age at first sample date | Age of the dog at the first sample date; [numeric] |
| diet | Control or test diet for a period of 1 year; [categorical] |
| sociability | Composed of the "dependency" component of the picture-viewing test and the "playfulness" component of the greeting and playing subtest. This variable indicates how the dog reacted to a stranger compared to the dog owner. [categorical] |
| trainability | Composed of four factors to measure how fast a dog performed a task and how attentive a dog was. The factors were: 1) latency to establish eye contact with an experimenter, 2) time spent at the gate when solving a detour-test, 3) time spent finding dropped food on the floor and 4) dependency of the dog towards its owner. [categorical] |
| boldness | Composed of two variables to evaluate the openness of the dog towards a novel person or environment. The variables measured were 1) reaction of the dog when greeted by a stranger and 2) the latency to find hidden food in the memory test. [categorical] |
| dependency | Composed of 1) dogs' reaction on separation from the owner in the separation subtest, 2) how independent of its owner the dog moved in the picture-viewing subtest and 3) how the dog reacted to its owner in the greeting and playing subtest–compared to a stranger. [categorical] |
| activity-independence | Composed of 1) the "activity/exploration" component in the exploration subtest and 2) the component of independence in the picture-viewing subtest. This variable indicates how independent a dog is acting of its owner. [categorical] |
| problem solving | Composed of 4 factors to assess the dogs' ability to solve different tasks. The factors were: 1) latency to success and amount of help seeking behavior in the detour subtest, 2) latency to act in the novel action subtest, 3) attentiveness in the attention subtest and 4) amount of time a dog manipulated a toy in a solvable and unsolvable task. [categorical] |

increasing trainability correlates with decreased TS, maintenance of TL, and lengthening of telomeres (Fig 1).

## Discussion

Surprisingly, the results of this study showed that only trainability had an effect on telomere change. Increasing trainability correlated with decreased TS, retention of TL, and even

**Table 2. Six best models after model selection based on the AICc.** Only models with a delta below 2 are shown.

| Intercept | trainability | boldness | dependency | problem solving | sociability | activity | Initial RTL | age | sex | body mass (kg) | diet | FCI-group | df | AICc | delta | weight |
|---|---|---|---|---|---|---|---|---|---|---|---|---|---|---|---|---|
| -24.12 | + | / | / | / | / | / | / | / | / | / | / | / | 4 | 532.2 | 0.00 | 0.087 |
| 11.86 | + | / | / | / | / | / | / | -3.65 | / | / | / | / | 5 | 532.9 | 0.66 | 0.062 |
| -17.74 | + | / | / | / | / | / | -3.77 | / | / | / | / | / | 5 | 532.9 | 0.71 | 0.061 |
| -4.37 | + | / | / | / | / | / | -3.99 | -3.86 | / | / | / | / | 6 | 533.4 | 1.20 | 0.048 |
| -28.35 | + | / | / | / | / | / | / | / | / | 0.22 | / | / | 5 | 533.9 | 1.69 | 0.037 |
| -25.53 | + | / | / | / | / | / | / | / | / | / | + | / | 5 | 534.1 | 1.87 | 0.034 |

The factors measured in the Modified Vienna Canine Cognitive Battery are shown with a gray background. The variable "activity-independence" is named "activity" in the table. "+" indicates that a categorical variable is included in the model, values indicate that a numeric variable is included in the model and "/" indicates that a variable is not included in the model.

**Table 3. Relative Variable Importance (RVI) of the five variables that are included in the six best models.** The results of the variables that have a higher RVI than 0.7 are printed bold. Trainability is included as a factor, comparing low trainability to medium and high trainability.

|  | Estimate | Standard Error | RVI |
|---|---|---|---|
| **Intercept** | -2.4 | 47.15 | / |
| **T_S_1st** | -4.7 | 4.72 | 0.38 |
| **trainability (medium)** | **5.78** | **7.87** | **1.00** |
| **trainability (high)** | **42.43** | **10.15** | - |
| **diet** | 3.49 | 6.12 | 0.27 |
| **body mass (kg)** | -0.18 | 0.52 | 0.26 |
| **age at sampling** | -0.003 | 0.005 | 0.39 |

lengthening of telomeres. The factor "Trainability" was composed of a shorter latency to establish eye contact with the experimenter and to find dropped food on the floor, spending less time at the gate while solving the detour and showing less dependency towards its owner.

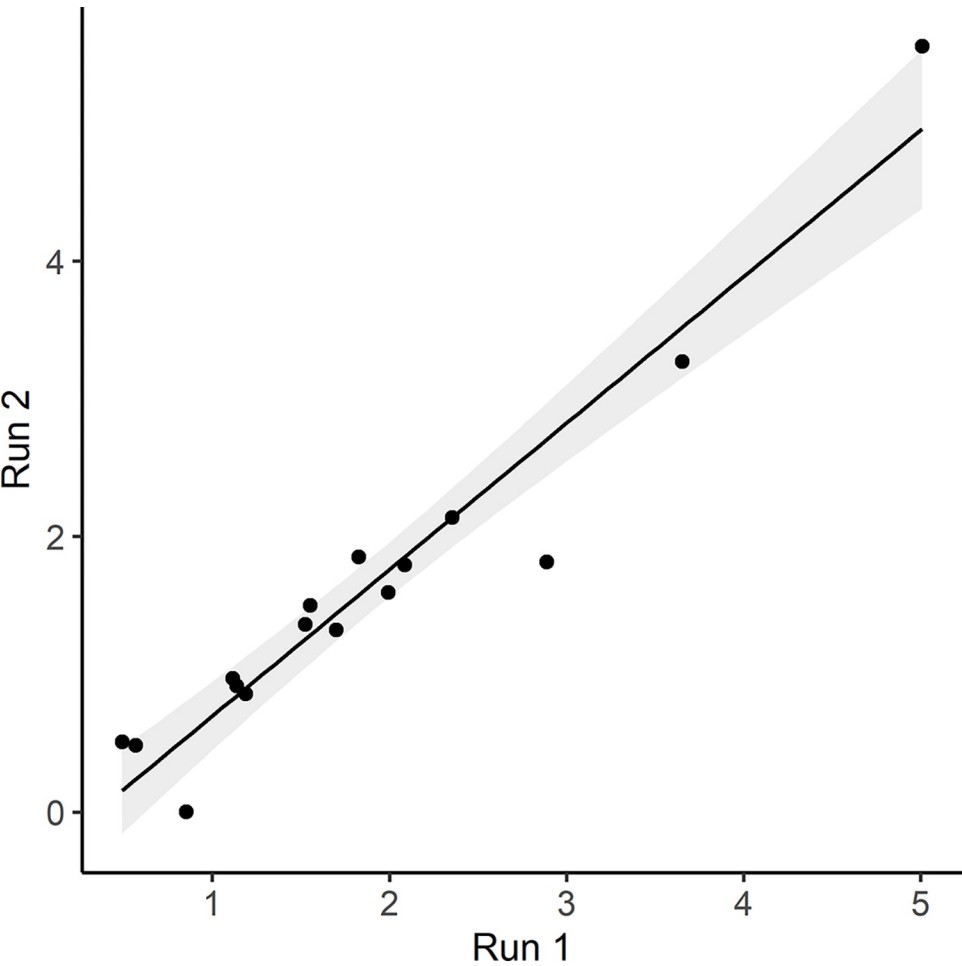

**Fig 2. Partial plot showing the influence of trainability on the percentage of telomere change (POTC) between the two sampling points.** The dots represent the raw data. Error-bars represent the predicted mean and 95% confidence intervals, always taking the other variables in the overall model into account. Negative values of telomere change indicate shortening of RTL over time, positive values are showing elongation of RTL over time. A trainability of 1 indicates low trainability, 2 medium and 3 high trainability.

Since these components measured how attentive dogs were and how fast they performed the tasks, the factor was named "Trainability" [53]. In other words, these results showed that the better their trainability during the study, the better it was for their telomere maintenance. Interestingly, [53] found that trainability showed no change with increasing age. Together with the data of this study, this may indicate that telomere dynamics in dogs are more dependent on a social trait, such as trainability, than on age. Even though age had a low RVI of 0.39 (Table 3), it was the second most important variable in this study. The indicated trend that telomeres were shorter in older individuals (Table 2) is in line with other literature in canines [43].

TS can be reversed by the activity of the enzyme telomerase or by alternative lengthening of telomeres [14–17] but how such pathways would be influenced by trainability are unclear.

Intelligent animals like dogs need mental and physical activity for their wellbeing [38–41]. Particularly hunting animals like wolves and dogs are capable of complex thinking and problem solving [68, 69]. They need these skills to survive and perform all standard tasks in daily life. Therefore, even after domestication, providing variety and regular problem-solving activities may be associated with positive effects for individuals with advanced cognitive abilities [34, 35]. This could mean that regular training has a positive effect on their general wellbeing and consequently on their telomere dynamics.

Dogs are very well adapted to the human lifestyle [41], since they have been one of the first species to be domesticated [42] and this process has been a driving force in the evolution of dog cognition [68, 70]. Dogs were domesticated and bred to work and live with humans, to obey commands, and to perform complex work requirements. The so-called working dogs are used in many fields, from search and rescue dogs to therapy dogs and many more. In order to meet the requirements in the respective work areas, extensive training, sometimes lasting years, is required, for which the dogs must be easily trainable.

[71] described that when evaluating military dogs prior to purchase, the dogs' current level of training and future trainability are assessed and extensive behavioral testing is conducted. The authors define specific behavioral problems associated with the discharge of dogs unsuitable for military training and analyze whether potential trainability issues could have been identified when evaluating these dogs before purchase. The results are incorporated into the future evaluation of dogs intended for purchase.

The main reason for service dogs being discharged is the lack of trainability due to inappropriate repertoire of behaviors, the most common being fear and aggression. Dogs that are more fearful and/or aggressive show lower trainability. Fear and aggression can lead to higher levels of stress, which could accelerate the shortening of telomeres. Working dogs being discharged from their service means not only the loss of valuable assets, but also the loss of the resources used to purchase, train and care for these dogs [71–73]. As most dogs show a preference for spending time with their owners, trainability could reflect the intensity of interaction between the dogs and their owners. Therefore, trainability would be an indicator for the dog's general wellbeing, which would influence the telomere dynamics. Such effects of social or psychological factors on telomeres have been described in different species (e.g., [20, 21, 74]). Interestingly, activity-independence, and dependency (two variables that explain the relationship between dog and owner) are not part of the best model and have no influence on telomere dynamics.

The cognitive parameters, boldness, problem solving, and sociability, did not show significant correlation, but were included in the best-models table. We suspect that this is because these social skills play a role in trainability. A dog must be bold to train, even under the possibility of making mistakes if necessary. It needs to solve problems when it follows commands. Since it is a human-animal interaction, the dog must be social while training together. Several

studies showed that behavioral and cognitive parameters are associated with telomere dynamics in humans but also animals [19–21, 74]. Concerning dogs, recent studies described that aging dogs show behavioral and cognitive changes [75, 76] and that some cognitive parameters, like Problem solving, Sociability, Boldness, and Dependency, show a linear decline with age [53].

Body weight did not show a significant correlation, but was included in the best-models table. At some level, weight and breed are coherent. Breed differences are known in behavior, genetics and telomere length [38, 43, 45–47] hence in lifespan and life expectancy.

There are many factors that can alter telomere length. This study showed that trainability is a factor that can influence telomere dynamics in dogs. This result could help veterinarians and researchers to develop new tools to improve the quality of life in aging dogs but it also showed that further research is needed in the areas of behavior, cognition and telomere dynamics.

## Supporting information

**S1 Table. All data used for the analyses.**
(XLSX)

**S2 Table. Dog breeds grouped into FCI groups based on the Fédération Cynologique Internationale (FCI) website (https://www.fci.be/en/).**
(PDF)

## Acknowledgments

We are grateful to Boglárka Bálint and Sarah Postner for their technical assistance and support in the laboratory. We thank Felix Knauer for his useful comments on the statistics and Renate Hengsberger for her help with the references and formatting of the manuscript.

Furthermore, we are grateful to the Messerli Institute and its staff, especially Julia Schoesswender, for their support throughout the project. We also thank Dr. Pakozdy and Dr. Leschnik for their collaboration, as well as the dog owners and dogs, for their participation and interest in the project.

## Author Contributions

**Conceptualization:** Julia Weixlbraun, Steve Smith, Franz Schwarzenberger, Franz Hoelzl.

**Data curation:** Julia Weixlbraun, Durga Chapagain, Franz Schwarzenberger, Franz Hoelzl.

**Formal analysis:** Julia Weixlbraun, Jessica Svea Cornils, Steve Smith, Franz Hoelzl.

**Investigation:** Julia Weixlbraun, Durga Chapagain, Franz Schwarzenberger, Franz Hoelzl.

**Methodology:** Jessica Svea Cornils, Steve Smith, Franz Hoelzl.

**Project administration:** Julia Weixlbraun, Franz Hoelzl.

**Validation:** Jessica Svea Cornils, Steve Smith, Franz Schwarzenberger, Franz Hoelzl.

**Visualization:** Julia Weixlbraun, Jessica Svea Cornils, Franz Hoelzl.

**Writing – original draft:** Julia Weixlbraun.

**Writing – review & editing:** Julia Weixlbraun, Durga Chapagain, Jessica Svea Cornils, Steve Smith, Franz Schwarzenberger, Franz Hoelzl.

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
