## [Decision Letter · Decision Letter 0]

29 Oct 2024

PONE-D-24-17700Impact of trainability on telomere dynamics of pet dogs (Canis lupus familiaris): An explorative study in aging dogsPLOS ONE

Dear Dr. Hoelzl,

Thank you for submitting your manuscript to PLOS ONE. After careful consideration, we feel that it has merit but does not fully meet PLOS ONE’s publication criteria as it currently stands. Therefore, we invite you to submit a revised version of the manuscript that addresses the points raised during the review process.

We look forward to receiving your revised manuscript.

Kind regards,

Gabriele Saretzki, PhD

Academic Editor

PLOS ONE

Journal Requirements:

Additional Editor Comments:

Please carefully address the comments from the 2 reviewers.

Reviewers' comments:

Reviewer's Responses to Questions

**Comments to the Author**

1. Is the manuscript technically sound, and do the data support the conclusions?

Reviewer #1: No

Reviewer #2: Yes

2. Has the statistical analysis been performed appropriately and rigorously? 

Reviewer #1: No

Reviewer #2: Yes

3. Have the authors made all data underlying the findings in their manuscript fully available?

Reviewer #1: Yes

Reviewer #2: Yes

4. Is the manuscript presented in an intelligible fashion and written in standard English?

Reviewer #1: No

Reviewer #2: Yes

5. Review Comments to the Author

Reviewer #1: This is a comparison of percentage telomere length change after approximately one year of two dietary treatments in dogs. I like the topic and appreciate the effort made. I think there is value in these data. I am not convinced of the appropriateness of the statistical analyses and interpretation. I encourage reconsideration of a revised manuscript that reports this work.

The major issues are 1) predictor description and quantification, and 2) the statistical analyses employed.

1) All evaluated predictors need to be explained in the Materials and Methods. At least the predictor ‘trainability’ was somewhat explained in the results section. I also note that trainability may be inherent in dogs (lines 299-300) or developed across time (line 292). If it is the latter, that may complicate the interpretation of results.

2) The statistical methodology used is pretty nonstandard. Why not just use analysis of variance that is interpretable by most researchers?

a. The modeled effects must be clarified and defined. For example, were the behavior assessments modeled as linear covariates or as categorical variables?—presentation of results suggests both. They are difficult to see, especially in Table 1. More description of the distributions of these values would be best.

b. AICc—I am not familiar with. Akaike’s information criterion (without the little c) is maybe useful for random structure determination but is less good for fixed effect determination. What would be wrong with just a simple model with evaluated effects? Analysis software (MUMin) should be in the Methods section.

c. FCI group is never defined.

d. Table 1—define table elements and numbers. Does this table mean that age was only important in results from two models? This is pretty inconsistent with telomere length investigations in many species and should be explained.

e. Lines 234-235 all possible additive models—this could mean a large number of analyses giving the opportunity to find something meaningful. Multiple testing issues may apply.

f. Lines 236-237 relative variable importance is not familiar to me and may benefit from additional description.

g. Table 2 All SE are larger than the estimates except for trainability (high). The number of dogs supporting this estimate would be good to know (looks like 14 of the 63 from figure 2). In figure 2, it looks like one sample strongly influences this and another 4 may be above the average POTC. This figure clarifies that this variable was categorical rather than a covariate.

h. lines 331-332 I think it is appropriate to be concerned about breed type since differential loss of telomeric sequences has been demonstrated for different breeds. Need to better describe the distribution of body weight in the text.

3) Minor issues

a. Not a fan of first person plural for scientific papers. I acknowledge that it is personal opinion and not critical.

b. Some issues with English, such as incorrect usage of singular and plural nouns and verbs, capitalization, and awkward sentence construction.

c. I recommend reporting results as past tense. Using present tense may imply that results are believed to accurately represent reality/truth. Scientific reports should probably be pretty conservative with respect to concluding truth.

d. Out of sequence description of names, etc., (e.g., Clever Dog lab), decreases readability.

e. Names of veterinarians and data collectors should be omitted from the text.

Reviewer #2: The article shows that repeated measurement of telomere length of cheek epithelial cells does not always show a decrease in length, and in some cases the length increases. Among many parameters studied, changes in telomere length are most strongly related to trainability. This is not a very well understood parameter. It is quickness of reaction, good appetite, cheerful disposition, increased motor activity, desire to play.

The disadvantages of the article include the following:

1. I have not found anywhere (except for the previous paper) an indication of the time between the first and second measurements.

2. There were 63 dogs involved in the study, but Fig. 1 shows only 16 points, Fig. 2 shows about 50 points. The authors should show the complete data.

3. Fig.2 shows 13 dots in the trainability level 3 group, but the original table shows only 12 dogs in this group. One is redundant.

4. I think the authors should show more raw data: it is very interesting, and most importantly it makes it easier to understand. It turns out that dogs have very different telomeres.

5. It is necessary to describe the essence of the term trainability in more detail. Perhaps the authors should discuss the anomalous behavior of this parameter from the first article. It is the only one that does not decrease with age! This is strange. And the result of this study may have something to do with it.

6. Classifying dogs based on questionnaires has a huge subjective component. Certainly, grasping the results of this paper, it should be repeated in a more objective way. It would be quite nice to measure blood cells in parallel.

I believe that the article will be worthy of publication in the Plos one journal after revision according to the remarks.

6. PLOS authors have the option to publish the peer review history of their article (what does this mean?). If published, this will include your full peer review and any attached files.

Reviewer #1: No

Reviewer #2: **Yes: **Dr. Yegor Yegorov

---

## [Author Response · Author response to Decision Letter 0]

13 Dec 2024

1) we made sure do meet all style requirements

2) we included captions for our supporting information files after the references in our manuscript

---

## [Decision Letter · Decision Letter 1]

27 Dec 2024

Impact of trainability on telomere dynamics of pet dogs (Canis lupus familiaris): An explorative study in aging dogs

PONE-D-24-17700R1

Dear Dr. Hoelzl,

We’re pleased to inform you that your manuscript has been judged scientifically suitable for publication and will be formally accepted for publication once it meets all outstanding technical requirements.

Kind regards,

Gabriele Saretzki, PhD

Academic Editor

PLOS ONE

Additional Editor Comments (optional):

The authors addressed all comments now.

Reviewers' comments:

Reviewer's Responses to Questions

**Comments to the Author**

1. If the authors have adequately addressed your comments raised in a previous round of review and you feel that this manuscript is now acceptable for publication, you may indicate that here to bypass the “Comments to the Author” section, enter your conflict of interest statement in the “Confidential to Editor” section, and submit your "Accept" recommendation.

Reviewer #2: All comments have been addressed

2. Is the manuscript technically sound, and do the data support the conclusions?

Reviewer #2: Yes

3. Has the statistical analysis been performed appropriately and rigorously? 

Reviewer #2: I Don't Know

4. Have the authors made all data underlying the findings in their manuscript fully available?

Reviewer #2: Yes

5. Is the manuscript presented in an intelligible fashion and written in standard English?

Reviewer #2: Yes

6. Review Comments to the Author

Reviewer #2: (No Response)

7. PLOS authors have the option to publish the peer review history of their article (what does this mean?). If published, this will include your full peer review and any attached files.

Reviewer #2: No

---

## [Editor Report · Acceptance letter]

2 Jan 2025

PONE-D-24-17700R1 

PLOS ONE

Dear Dr. Hoelzl, 

I'm pleased to inform you that your manuscript has been deemed suitable for publication in PLOS ONE. Congratulations! Your manuscript is now being handed over to our production team.

Kind regards, 

on behalf of

Dr. Gabriele Saretzki 

Academic Editor

PLOS ONE